# Redox Targets for Phosphine–Boranes

Yonatan Morocz [1,2] , Rachel E. Greben [1,3,4] and Leonard A. Levin [1,5,*]

1    Department of Ophthalmology and Visual Sciences, McGill University, Montreal, QC H4A 3S5, Canada
2    Department of Chemistry, McGill University, Montreal, QC H3A 0G4, Canada
3    Department of Pharmacology & Therapeutics, McGill University, Montreal, QC H3A 1A3, Canada
4    Temerty Faculty of Medicine, University of Toronto, Toronto, ON M5S 1A8, Canada
5    Department of Neurology and Neurosurgery, McGill University, Montreal, QC H3A 1A1, Canada
*    Correspondence: leonard.levin@mcgill.ca

**Abstract:** Understanding the complex mechanisms underlying redox-mediated biological processes is a fundamental pillar of cellular biology. We describe the identification and quantification of disulfide formation and reduction in response to phosphine–borane complexes. We illustrate the specific cysteine reduction effects of the novel phosphine–borane complex bis(3-propionic acid methyl ester) phenylphosphine–borane complex (PB1) on cultured 661W cells. A total of 1073 unique protein fragments from 628 unique proteins were identified and quantified, of which 13 were found to be statistically significant in comparison to control cells. Among the 13 identified proteins were Notch1, HDAC1, UBA1, USP7, and subunits L4 and L7 of the 60S ribosomal subunit, all of which are involved in redox or cell death-associated pathways. Leveraging the ability of tandem mass tagging mass spectrometry to provide quantitative data in an exploratory manner provides insight into the effect PB1 and other phosphine–borane compounds may have on the cysteine redoxome.

**Keywords:** boron; phosphine–borane complexes; mass spectrometry; cysteine; tandem mass tagging; redox; disulfide





## 1. Introduction

Borane-containing compounds have been used in medical treatments to prevent viral or bacterial replication, treat cancer, and mitigate neurodegeneration [1–3]. Phosphine–borane (PB) complexes have been shown to selectively reduce disulfide bonds, delaying RGC death in cultures of axotomized RGCs [4,5], optic nerve transection [6,7], and an experimental ocular hypertension model in rats [7]. The positively charged borane zwitterion prevents exposed phosphine compounds from cleaving disulfide bonds indiscriminately and provides air and moisture stability [5,8,9]. In the presence of amine groups, phosphine–borane groups are deprotected through an $S_N2$ nucleophilic substitution reaction [7,9]. Once deboronated, the phosphine compounds are both membrane permeable and able to cleave disulfide bonds [8,10]. As demonstrated by Dmitrenko et al., these compounds are capable of cleaving disulfide bonds through an $S_N2$ nucleophilic substitution reaction where the exposed phosphorus acts as a nucleophile [11]. This behavior was further characterized using intracellular dithiol reporters to validate that phosphine borane compounds are able to cleave intracellular disulfide bonds [12]. Prior work illustrated the concentration-dependence of PB1, showing RGC rescue in vitro after axon transection at 100 pM, with increased levels of cell rescue at 1 nM followed by a plateau up to 100 µM [5].

Reducing and oxidizing conditions have critical effects on biologically relevant molecules, signaling pathways, and cellular functions. Changes in the redox environments within cells can dramatically affect the interactions of many proteins through the modification of enzymes such as kinases and the indirect modulation of cysteine-rich proteins such as glutathione S-transferase [11]. Certain reactive oxygen species (ROS) can act as signal transduction messengers and transcription factors, altering gene expression based on

physiological oxidant–antioxidant homeostasis [13]. Neurons specifically are very sensitive to their redox environment, and ROS signaling can trigger somatic cell death after injury. Many drugs are therapeutically active via a mechanism of action that involves inhibiting reactive oxygen species (ROS) formation or activating endogenous antioxidant defense systems [14,15].

Given that redox-active PBs have neuroprotective effects in models of axonal damage that may be mediated by their disulfide-cleavage activity, we sought to determine the specific disulfide targets that are reduced by PBs in a disease state. We previously demonstrated that a simplified redox proteomic method using non-reducing/reducing 2-dimensional gel electrophoresis ("diagonal gels") could be used to identify arrestin as a target for tris(2-carboxyethyl)phosphine (TCEP) [16]. Unfortunately, the diagonal gel technique is less successful in identifying low-abundance targets because of its lower sensitivity.

We describe a new method for identifying disulfide formation and reduction in protein targets of redox-active drugs and apply it to studying the role of phosphine–borane (PB) complexes in preventing neuronal death. The resultant solution leverages the sensitivity of mass spectrometry and iodoTMT labeling of free cysteines in various states of reduction to identify the disulfide targets. In contrast to existing methods, which measure a more global redox state or only measure the redox state of predetermined proteins, this method measures specific disulfide reduction targets across the entire proteome of the desired cells or tissue. Additionally, the multiplexed ability of the method allows for a quantitative comparison of different agents within a single mass spectrometer run. These features make the proposed method ideal for this exploratory study on the reducing behavior of phosphine–borane complexes and the elucidation of the cellular mechanics underlying their neuroprotective effects.

## 2. Results

### 2.1. Global Effects of Borane-Containing and Non-Borane-Containing Reducing Drugs on the Cysteines of 661W Proteins

A total of 3317 separate protein fragments were identified as protein spectrum matches (PSMs) over three labeling runs carried out according to the scheme shown in Figure 1. Of these, 2897 PSMs were identified in all the PB1, TCEP, and control samples. Once multiple hits from single PSMs across experiments were combined into single entries, there remained 1073 unique PSMs from 628 unique proteins.

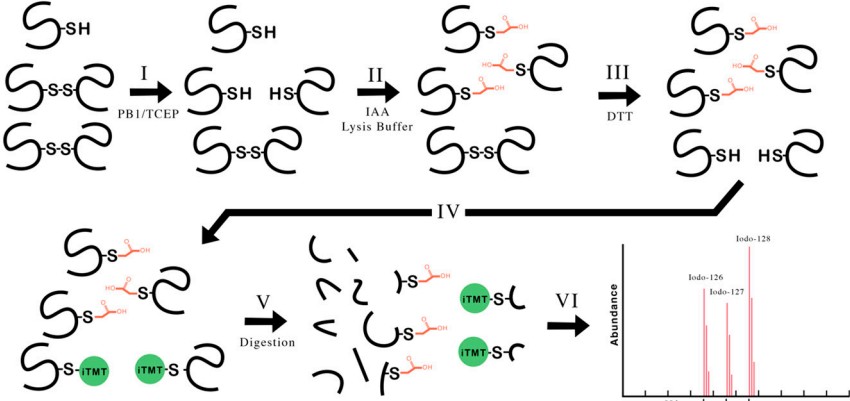

**Figure 1.** Scheme for redox proteomics of disulfides exposed to reducing conditions (PB1 and TCEP). (I) The disulfides present are exposed to the reducing agents, resulting in the breaking of specific disulfides. (II) The original sulfhydryls and those formed by the breaking of disulfides are then alkylated by IAA to prevent further labeling. (III) DTT treatment reduces the remaining disulfides, resulting in free sulfhydryls. (IV) The newly formed sulfhydryls are now labeled with the iodoTMTsixplex reagents. (V) Finally, the proteins are digested, and the labeled PSMs are extracted with iodoTMT enrichment resin. (VI) The labeled PSMs are analyzed using mass spectrometry.

However not every PSM was present in all three of these trials, and, if only those are considered, then only 139 unique PSMs and 104 unique proteins remained. However, by limiting the analysis to only those PSMs detected in all three samples, the analysis overlooked certain PSMs that had such low abundances in the cells that they were naturally close to the threshold of the instrument. These PSMs may have been over the threshold in one trial, and just slightly below in another.

The PB1 and TCEP abundances for each PSM entry were divided by the control abundance for that PSM to yield a ratio showing its relative abundance. The ratios generated with this process for both PB1 and TCEP were centered around 1, illustrating that the majority of the PSMs were not altered from their original states.

In general, the ratios deviated further in the positive direction than the negative, with a decrease of over 20% (ratio < 0.8) only observed in PSMs not detected in all three trials. There were slightly higher positive deviations up to 40% detected in the PSMs in all three trials, as shown in Figure 2.

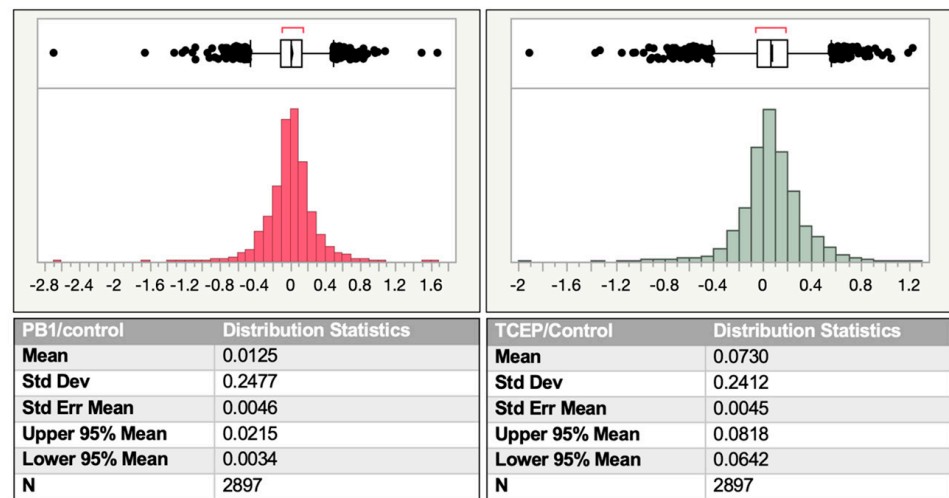

| PB1/control | Distribution Statistics | | TCEP/Control | Distribution Statistics |
|---|---|---|---|---|
| Mean | 0.0125 | | Mean | 0.0730 |
| Std Dev | 0.2477 | | Std Dev | 0.2412 |
| Std Err Mean | 0.0046 | | Std Err Mean | 0.0045 |
| Upper 95% Mean | 0.0215 | | Upper 95% Mean | 0.0818 |
| Lower 95% Mean | 0.0034 | | Lower 95% Mean | 0.0642 |
| N | 2897 | | N | 2897 |

**Figure 2.** Distribution observed in log ratios of PB1/control (**left**) and TCEP/control (**right**). (**Top**) Distribution histogram for log ratios. (**Bottom**) Statistics for log-ratio distribution.

## 2.2. Effects of PB1 on the Cysteines Present in 661W Cell Proteins

PSMs that passed the significance threshold based on a multiple-comparisons analysis using analysis of means (ANOM) and Nelson's h are displayed in Table 1 along with their ratios and parent proteins. For ease of analysis and to demonstrate the separation of the significant entries from the rest, the output of the multiple-comparisons analysis was visualized in a Central Park plot, shown in Figure 3, with the genomic location as the x-axis. The Central Park Plot identified 13 distinctly labeled PSMs that had statistically significant PB1/control ratios.

## 2.3. Biological Relevance of Identified Targets

Ubiquitin carboxyl-terminal hydrolase 7 (USP7) was labeled at cysteine 300 with a PB1/control ratio of 0.341 (i.e., the cysteine was more strongly labeled in the control compared to PB1). Ubiquitin-like modifier activating enzyme 1 (UBA1), on the other hand, was labeled at cysteine 906 with a PB1/control ratio of 1.87 shown in Figure S3 (i.e., PB1 incubation indirectly resulting in disulfide formation or some other cysteine modification). The opposite values for the PB1/control ratio make sense mechanistically given that USP7 is involved in deubiquitylation [17], while UBA1 is involved in the counter-process of ubiquitylation [18]}. Both proteins were labeled at isolated cysteines that were not located near the surface of the protein. This is particularly interesting in the context of recent work describing redox-sensitive ubiquitylation via a mechanism involving the reduction of an intermolecular disulfide bond between UBA1 and E2 [19]. Similarly, Doris and colleagues

found that interactions between UBA1 and specific E2 proteins can be modulated by sequestering catalytic cysteines with oxidative inactivation [20]. The nature of how these labeled cysteines relate to the functions of UBA1 and USP7 in the present study is unknown and warrants further investigation.

**Table 1.** Abundance ratios of significant PSMs for PB1/control.

| Protein | Protein Accession Code | Annotated Sequence | Positions in Proteins | Abundance Ratio (Sample/ Control) | $p$ |
|---|---|---|---|---|---|
| Neurogenic locus notch homolog protein 1 | Q01705 | [R].CEGDVNEcLSNPcDPR.[G] | [1264–1279] | 0.068 | $5.0 \times 10^{-11}$ |
| Thrombospondin-4 | Q9Z1T2 | [K].DGIGDEcDDDDDNDGIPDLV PPGPDNcR.[L] | [659–686] | 0.330 | $1.1 \times 10^{-3}$ |
| Histone deacetylase 1 | O09106 | [R].FNVGEDcPVFDGLFEFcQLSTGG SVASAVK.[L] | [94–123] | 0.340 | $2.1 \times 10^{-3}$ |
| Ubiquitin carboxyl-terminal hydrolase 7 | Q6A4J8 | [K].SFGWETLDSFMQHDVQELcR.[V] | [283–302] | 0.341 | $2.2 \times 10^{-3}$ |
| Elongation factor 1-alpha 1 | P10126 | [K].NMITGTSQADcAVLIVAAGVGE FEAGISK.[N] | [101–129] | 0.491 | $2.8 \times 10^{-6}$ |
| Transmembrane emp24 domain-containing protein 9 | Q99KF1 | [KR].cFIEEIPDETmVIGNYR.[T] | [49–65] | 0.887 | $3.4 \times 10^{-2}$ |
| Protein S100-A11 | P50543 | [R].cIESLIAVFQK.[Y] | [8–18] | 1.307 | $2.9 \times 10^{-2}$ |
| Elongation factor 1-gamma | Q9D8N0 | [R].WFLTcINQPQFR.[A] | [190–201] | 1.712 | $4.3 \times 10^{-4}$ |
| Ubiquitin-like modifier-activating enzyme 1 | Q02053 | [K].IIPAIATTTAAVVGLVcLELYK.[V] | [890–911] | 1.873 | $5.3 \times 10^{-3}$ |
| DNA dC->dU-editing enzyme APOBEC-3 | Q99J72 | [R].LYNVQDPETQQNLcR.[L] | [142–156] | 2.558 | $3.6 \times 10^{-9}$ |
| 116 kDa U5 small nuclear ribonucleoprotein component | O08810 | [K].VEESGEHVILGTGELYLDcVM HDLR.[K] | [620–644] | 2.615 | $3.6 \times 10^{-2}$ |
| 60S ribosomal protein L4 | Q9D8E6 | [R].FcIWTESAFR.[K] | [249–258] | 3.012 | $8.5 \times 10^{-10}$ |
| 60S ribosomal protein L7 | P14148 | [K].FGIIcMEDLIHEIYTVGK.[R] | [204–221] | 4.484 | $1.0 \times 10^{-7}$ |

An intermolecular disulfide bond was labeled between the L4 and L7 proteins in the 60S ribosomal subunit shown in Figure S4 [21]. L4 was labeled on cysteine 250 with a ratio of 3.01, while L7 was labeled on cysteine 208 with a ratio of 4.48, within 0.2 log of each other. The similarity in the ratios results from the two cysteines originating from the same disulfide bond. Factors such as the accessibility of the cysteines to the labeling agent, the alkylating agent IAA, and less likely, the reformation of a disulfide bond with other molecules or proteins could account for this difference in the context of a high-sensitivity redox proteomic technique. Oxidative stress and ROS can modify rRNA within ribosomes, thereby inhibiting the biosynthesis of proteins [20]. Additionally, subjecting ribosomal proteins to different redox environments acting on specific cysteines that serve as redox sensors was shown to influence several aspects of translation [22]. The intermolecular disulfide between the L4 and L7 subunits of the 60S ribosomal subunit labeled in the present study could have a translation-modifying function relevant to the neuroprotection observed with PB1 [5,7,12,22].

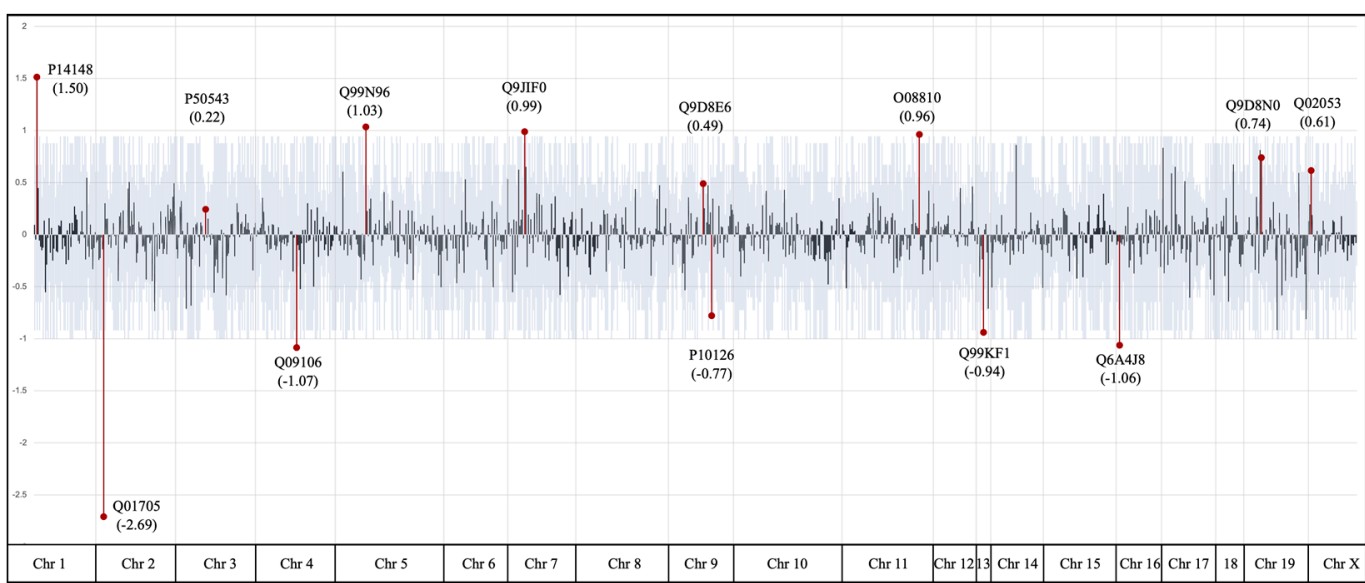

**Figure 3.** Central Park plot for PB1/control entries. Entries are ordered on the x-axis according to their genomic location. The y-axis is the log of the PB1/control ratio. Statistically significant points, corrected for multiplicity, are indicated with a red dot. The associated PB1/control ratio and the protein accession code is listed adjacent to each significant point.

### 2.4. Comparison between PB1- and TCEP-Mediated Reduction

The purpose of adding the TCEP trial to the experiment was to compare the specific effects of PB1 to the much more general effects of a well-known laboratory-reducing agent. Additionally, this served a secondary purpose of illustrating the method's ability to differentiate between conditions. Unlike borane-containing PB1, TCEP is a non-cell permeating reducing agent, allowing for a comparison between the effects of intracellular and extracellular reducing agents. Important differences between PB1 and TCEP are illustrated in Table S1 and Figure S1. The most relevant PSMs were identified as those with a large difference between the PB1 and TCEP measurement, statistically significant PSMs are displayed in Table 2. These are the fragments where the differential reduction effects of PB1 would most likely be found. The log distribution of the TCEP/control ratios was centered around 0, similar to the distribution of the PB1/control ratios shown in Figure 2.

**Table 2.** Abundance ratios of significant PSMs for TCEP/control.

| Protein | Protein Accession Code | Annotated Sequence | Positions in Proteins | Abundance Ratio (Sample/ Control) | $p$ |
|---|---|---|---|---|---|
| Neurogenic locus notch homolog protein 1 | Q01705 | [R].CEGDVNEcLSNPcDPR.[G] | [1264–1279] | 0.149 | $5.0 \times 10^{-11}$ |
| Collagen alpha-1(I) chain | P11087 | [K].SGEYWIDPNQGcNLDAIK.[V] | [1260–1277] | 0.629 | $2.1 \times 10^{-2}$ |
| Elongation factor 1-alpha 1 | P10126 | [K].NMITGTSQADcAVLIVAAGVGEFEAGISK] | [101–129] | 0.661 | $7.9 \times 10^{-3}$ |
| Phosphoglycerate kinase 1 | P09411 | [K].TGQATVASGIPAGWMGLDcGTESSK.[K] | [298–322] | 0.736 | $8.5 \times 10^{-3}$ |
| Elongation factor 2 | P58252 | [R].LmEPIYLVEIQcPEQVVGGIYGVLNR.[K] | [740–765] | 0.886 | $6.2 \times 10^{-5}$ |

**Table 2.** *Cont.*

| Protein | Protein Accession Code | Annotated Sequence | Positions in Proteins | Abundance Ratio (Sample/ Control) | $p$ |
|---|---|---|---|---|---|
| Transmembrane emp24 domain-containing protein 9 | Q99KF1 | [KR].cFIEEIPDETmVIGNYR.[T] | [49–65] | 1.046 | $4.4 \times 10^{-2}$ |
| DNA dC->dU-editing enzyme APOBEC-3 | Q99J72 | [R].LYNVQDPETQQNLcR.[L] | [142–156] | 2.278 | $10.0 \times 10^{-6}$ |

## 3. Discussion

The overall goal of these experiments was to investigate potential protein targets by which novel phosphine–borane complexes might exert their neuroprotective effect [7]. Chemical reduction of oxidized sulfhydryls is neuroprotective in RGC injury, and the phosphine–borane complexes were designed with this in mind. Determining which sulfhydryls are reduced by phosphine–borane complexes, as exemplified by PB1 in this study, is an important step in identifying their mechanism of action. The identification of specific targets also allows for the development of more specific and effective molecules.

The majority of the analyzed PSMs showed ratios of PB1/control and TCEP/control centered around 1, indicating that neither drug caused specific changes in the protein disulfide redox profile. Because of the large number of proteins that had changes in varying degrees, we used statistical techniques to correct for multiple comparisons. Thirteen proteins were found with significant ratios, of which nine had changes in a factor of 2 higher or lower.

The current method was developed contemporaneously with the OxiTMT method of Shakir et al. [23], with the main difference being our simultaneous incubation with IAA at the time of cell lysis instead of the delayed incubation with IAA >1 h after cell lysis and washing with the other method. This could result in decreased sensitivity and specificity to detect the effects of a short exposure to a redox-active drug. Another difference is our use of a single iodoTMT label instead of two per condition as in Shakir et al., allowing for twice as many potential multiplexed conditions.

To understand the role of the identified cysteines within their parent proteins, the location within the larger protein and the local environment must be taken into consideration. From the initial inspection of the crystal structures in the identified proteins, the labeled cysteines tended to reside deeper within the protein and not on the surface. Five of the thirteen proteins listed above contained hits on isolated cysteines, while the other eight proteins exhibited potential intermolecular or intramolecular disulfide interactions. An isolated cysteine can still be labeled under the labeling protocol, especially when located deeper inside a protein, because the IAA alkylation step may not be able to protect it before labeling it with the iodoTMT reagent.

Identifying how different cysteines react to reducing agents is crucial to understanding these redox proteomic data. An examination of the structures of protein hits highlighted three distinct situations. In the first, the label corresponded to a single cysteine, isolated from other cysteines. Using other protein tagging protocols, it is possible to observe an inefficiency of alkylation in the middle of proteins. Detecting the difference between a reduced disulfide and a sulfhydryl that underwent incomplete alkylation is difficult. This or the presence of homodimers or heterodimers would most likely be the cause of isolated cysteine labeling. In the second situation, the structure showed two adjacent cysteines but without a disulfide bond. This could also be due to the presence of a homodimer, poor resolution, and/or preparation of the published crystal structures. Preparing these proteins for X-ray crystallography may by itself be responsible for reducing the disulfide.

Although PB1 reduces disulfides [12], the data obtained with this method demonstrated some proteins where there was no identifiable disulfide being reduced, and others

where there was an increase in disulfide formation, whether protein–protein or mixed. An example of how this could occur is the case of protein disulfide isomerase (PDI). During oxidative stress, glutathione (GSH) can react with a protein thiol to form a protein-mixed disulfide (protein–glutathione disulfide; PSSG) catalyzed by PDI [24,25]. This protein-mixed disulfide may then be exchanged for another protein thiol, resulting in a protein disulfide [26,27]. This reaction is typically catalyzed by PDI located in the endoplasmic reticulum. The cells in this experiment were lysed prior to labeling, potentially enabling PDI to catalyze the formation of disulfide bonds.

The majority of significant proteins differed between the two drugs, with only four statistically significant PSMs shared between PB1 and TCEP. Specifically, Notch homolog protein 1 and elongation factor 1-alpha showed decreased labeling in both TCEP and PB1 compared to the control. The reverse also occurred with the enzyme APOBEC-3, which was increased in both TCEP and PB1 experiments. Lastly, transmembrane protein 9 was slightly elevated compared to the control for TCEP and slightly below the control for PB1. Thrombospondin, histone deacetylase, ubiquitin hydrolase, protein S100, and elongation factor 1-gamma saw a significant abundance reduction with PB1 and not with TCEP.

In earlier concentration-dependence studies, PB1 showed neuroprotection of axonally transected RGCs in vitro starting at 100 pM with increased protection at 10 nM and above. Understanding the mechanism by which picomolar or nanomolar therapeutic doses enact neuroprotective effects could provide valuable insights for future drug discovery [7,12]. Two likely mechanisms underlying the activity at low concentrations are (1) the presence of the borane to protect the phosphine from unwanted oxidative side-reactions, with the borane removed by nearby amines [9] and (2) the methyl esters undergoing hydrolysis by intracellular esterases, thereby preventing the now polar molecule from exiting the cell.

Currently, the mechanisms by which the disulfide reduction of one or more targets identified within this paper could translate to a downstream effect are unclear. For example, multiple proteins are associated with retinal neuronal survival (photoreceptor or RGC) but may be downstream or upstream of neuroprotection from disulfide reduction [28–31]. PB1 induces ERK1/2 activation in experimental models of optic nerve injury, but how that process is involved in its neuroprotective activity is still under investigation [7]. We speculate that the reduction of Notch1 is a candidate for involvement in neuroprotection due to its regulation of the ERK1 and ERK2 cascades [32], which would tie a known redox effect of a phosphine–borane complex to a known biological effect.

Similarly, both histone deacetylase 1 and elongation factor 1 are related to the processes of neuroprotection, which have been documented with PB1. Elongation factor 1 regulates the metabolic pathways responsible for the processing of reactive oxygen species in cellular detoxification Sasikumar, 2012 #10664}. Modification of HDAC1 cysteines coincides with increased transcription of HDAC1-repressed genes such as heme oxygenase 1, heat shock proteins, and gadd45 [33].

Some limitations of this study should be noted. First, only a single short reduction duration was used, so any events occurring earlier or later in the reduction were not analyzed. Longer incubation times have been used in studies on cytoprotection with these drugs [5,34], but to observe more mechanistic immediate effects, the incubation was limited to one hour. Second, only one concentration of the reducing agents was tested, and the experiments were performed with only two compounds (PB1 and TCEP). Third, some PSM entries appeared in only one or two of the PB1, TCEP, or control channels, while others appeared in only one or two of the experimental replicates. All entries with missing channels had low (less than 6) abundances, likely due to their proximity to the instrument's sensitivity threshold. However, when comparing multiple experimental replicates, there were occasionally high-abundance entries that appeared in only one replicate, which likely reflects biological variability.

The motivation behind this investigation was to elucidate the mechanisms by which novel phosphine–borane complexes, in this case, PB1, achieve their neuroprotective effects. Among the targets identified, several proteins participate in cellular processes linked to

neurodegeneration. These include the labeling of two inverse ubiquitin-related proteins USP7 and UBA1, multiple proteins in the 60S ribosomal subunit, elongation factor 1, histone deacetylase 1, and Notch1. The specific cysteine residues undergoing redox modulation with PB1 identified here represent an important step in understanding the mechanism of action of other phosphine–borane complexes and related redox-active compounds.

## 4. Materials and Methods

### 4.1. Synthesis of PB1

PB1 was synthesized according to our published methods [5] at the Keck-University of Wisconsin Comprehensive Cancer Center Small Molecule Screening Facility (Madison, WI, USA). Briefly, phenylphosphine was dissolved in acetonitrile under argon, potassium hydroxide was added, and the resulting solution was cooled to 0 °C. Methyl acrylate was added at a rate that maintained the reaction temperature below 35 °C, and when complete, it was heated at 50 °C for 8 h and then washed with brine twice. The organic layer was dried over MgSO$_4$, filtered, and concentrated under a vacuum. The residue was purified using distillation and isolated as a clear liquid (81% yield). It was then dissolved in dry THF under argon, cooled to 0 °C, and borane-THF was slowly added. The reaction was stirred at 0 °C for 45 min and then at room temperature for an additional 1.5 h. The solvent was removed under reduced pressure, and the residue was purified using flash chromatography (silica gel, 80% *v*/*v* methylene chloride in hexanes). PB1 was isolated as a clear oil (35% yield, purity > 99% using HPLC).

### 4.2. Protocol to Identify Disulfide Bonds Associated with Phosphine–Borane Complex Activity

Disulfides present in vitro at the beginning of the example underwent the following transformations: As the cells were exposed to a redox-active agent, the disulfides in question were either reduced or not reduced. Next, the proteins were subjected to alkylating iodoacetamide (IAA), which protected any sulfhydryls present, regardless of whether they were present in the protein prior to the reducing step or were formed due to the reduction of a disulfide. Then, dithiothreitol (DTT) was added, which reduced all remaining disulfides. At this point, the only sulfhydryls present were the ones newly formed, with all sulfhydryls formed in the earlier reduction step still protected. The iodoTMT 6plex reagent was then added, labeling all the free sulfhydryls. However, if a disulfide was reduced by a drug or condition, then it would not be labeled with iodoTMT, because it would have already been protected with IAA. At this point, enrichment occurred, which removed all protein fragments not containing a label. A disulfide that was completely reduced with the drug or condition would not show up in either of those channels. The measured abundances were then converted into ratios to better illustrate the levels relative to a control.

The methods below describe the application of this protocol to determine targets of the phosphine–borane complex PB1 in a cone photoreceptor neuronal cell line.

### 4.3. Exposure of 661W Cells to Borane-Containing or Non-Borane-Containing Reducing Drugs

First, 661W cells were cultured in DMEM in T-25 flasks plated at 7500 cells/cm$^2$ 2 days prior to harvest. On the day of harvest, 5 μL aliquots of 100 mM PB1 and TCEP or DMEM were added to the cell growth solution to reach a final concentration of 100 μM. This concentration was shown to be non-toxic in a previous experiment. They were then incubated at 37 °C for 1 h before being moved into the harvest and labeling protocols.

### 4.4. Cell Harvest Protocol

Once the desired concentration of cells was achieved in the growth flask, the reducing agent was added, and the sample was incubated at 37 °C for 1 h (Figure 1). The DMEM was then aspirated, leaving behind the cells. The cells were then washed with 5 mL of 37 °C PBS. An initial 1 mL lysis buffer solution of 1% IGEPAL, 0.5% sodium deoxycholate, and 0.1% SDS was mixed with 1 mL of RIPA buffer, 37 mg of IAA, and a protease inhibitor tablet 30 min prior to use. The PBS was then aspirated, and 500 μL of the newly prepared

lysis buffer was added. The flask was then scraped using rubber flask scrapers until all the material in the flask was solubilized. The sample was then left on ice in the lysis buffer for 30 min. At this point, the sample was centrifuged at 14,000× $g$ for 10 min at 4 °C. The supernatant was collected for further processing, and the pellet was discarded.

### 4.5. Labeling Protocol

The protein concentration of the supernatant was obtained using a BCA assay. Then, 1.5 mL of −20 °C chilled acetone was added to each sample to precipitate the proteins. The acetone samples were then chilled at −20 °C for one hour before centrifugation at 10,000× $g$ for 10 min. The acetone was decanted, and the pellet was allowed to dry or undergo lyophilization. The protein pellet was then suspended in 200 μL of a denaturing buffer that was supplemented with DTT for a final DTT concentration of 20 mM (denaturing buffer, 6 M urea, 100 mM Tris, protease inhibitor tablet, 20 mM DTT) At this point, a second acetone precipitation was necessary because the DTT in the denaturing buffer deactivates iodoTMT reagents. Thus, 1.2 mL of −20 °C chilled acetone was added to each sample, and the samples were allowed to rest at −20 °C for 4 h or up to overnight in order to fully precipitate the proteins. This step was followed by centrifugation at 10,000× $g$ at 4 °C for 10 min. The acetone was then decanted, and the sample was dried using a lyophilizer. Each label was solubilized in 10 μL of LC/MS grade methanol and then centrifuged at 1500× $g$ for 1 min. The protein pellets from the acetone precipitation step were solubilized in 100 μL of denaturing buffer without the additional DTT used in the previous denaturing buffer step. Next, 10 μL of each label was added to each respective sample. The samples were then allowed to react for 1 h at 37 °C while being protected from light. The labeling reaction was quenched by adding 4 μL of 0.5 M DTT stock to each sample. At this point, equal volumes of each sample were combined into a final vial. A third acetone precipitation followed, carried out in the same manner as the previous. The pellet was then resolubilized in 300 μL of 50 mM ammonium bicarbonate. Trypsin was added to the vial in a 1/25 ratio of (weight of trypsin)/(weight of protein). The samples were allowed to digest overnight at 37 °C. Then, 400 μL of Thermo Fisher iodoTMT enrichment resin was added to a Thermo Fisher separation vial column. The resin was washed three times with 400 μL or one column volume of 1x TBS. The protein pellet from the trypsin step was then resuspended in 400 μL of the 1x TBS and added into the column containing the anti-TMT resin. The suspension was then set on an Eppendorf rotator at room temperature for 2 h. The resin was then washed an additional 3 times with 1x TBS followed by 3 times with distilled water for 5 min each time. Finally, the column is washed 4 times with 400 μL of TMT elution buffer. The pooled eluate was then evaporated in a lyophilizer to form a pellet. Finally, the sample was lyophilized and stored at −20 °C until ready to run. The samples were resuspended in 25 μL of 5% ACN + 0.1% formic acid and injected in a 1–5 μL aliquot directly into the LC-MS/MS system.

### 4.6. Mass Spectrometer System and Operation

μHPLC-MS/MS Parameters: IodoTMT-labeled and affinity-purified peptides were directly re-solubilized in 25 μL of 5% ACN + 0.1% formic acid. Then, 2 μg of peptides were loaded onto a Thermo Acclaim PepMap (Thermo, 75 μM ID × 2 cm C18 3 μM beads) precolumn and then onto an Acclaim PepMap EASY-Spray analytical column (Thermo Fisher, Waltham, MA, USA, 75 μM × 50 cm with 2 μM C18 beads) separation using a Dionex Ultimate 3000 UHPLC at 220 nL/min with a gradient of 2–35% organic (0.1% formic acid in acetonitrile) over 3 h. Peptides were analyzed using a Thermo Fisher Orbitrap Fusion mass spectrometer operating at 120,000 resolution (full-width at half-maximum) with linear ion trap sequencing of all peptides with a charge of 2+ or greater with reporter ion-neutral loss triggered HCD MS3 scans for improved reporter ion statistics. Mass spectrometry was carried out with assistance from the proteomics platform at the Research Institute of the McGill University Health Center.

### 4.7. Mass Spectrometer Data Processing and Analysis

Processing of the mass spectrometer data was performed with Proteome Discoverer 2.3 (Thermo Fisher, Waltham, MA, USA). PDBs were accessed from the Research Collaboratory for Structural Bioinformatics (RCSB) Protein Data Bank (PDB).

### 4.8. Statistics

Given the nature of the method, it was necessary to impose stringent parameters when deciding which hits were deserving of further attention. The abundances obtained for TCEP and PB1 for each PSM were divided by the control abundance to provide ratios. The log of those ratios was subjected to a fit model with JMP statistics software (SAS; Cary, NC, USA) followed by a multiple-comparisons analysis performed using analysis of means (ANOM) with Nelson's h statistic for constructing decision limits [35].

### 4.9. Toxicity Assessment

Cells were plated in a 24-well plate at a concentration of 200, 500, 1000, and 2000 cells/cm$^2$ in 200 μL of DMEM and imaged at 10× and 20× over three days. They were exposed to 10, 30, and 100 μM PB1 and TCEP concentrations, respectively. By the third day, cells in the highest seeding concentrations started to die due to overconfluency, but no visually apparent adverse effects of the reducing agents were noted at any concentration.

### 4.10. Consistency of Parallel Cultures

To assess whether parallel cultures were consistent, 661W cells were grown in a 5 mL flask until confluency and then harvested using the cell harvesting protocol described above. A protein assay was then performed on a 5× and 10× dilution of the protein mixture to accurately gauge the concentration. The newly found concentration was then used to determine the volume needed for running the gel. Then, 1–10 μg of protein sample was added to each well with a Coomassie blue stain. The gel was then run at 120 V until the stain neared the end of the gel. At this point, the gel was stained for 15 h with 40% ethanol, 10% acetic acid, and 0.1% Coomassie R-250. Following this, the gel was de-stained twice for 30 min each time with 10% ethanol and 7.5% acetic acid. The volume used for each staining step was just enough to fully cover the gel and allow its proper movement. The gel was then imaged using a vis spectrum imager and a Li-Cor Odyssey gel scanner at 700 nm.

### 4.11. Detection of iodoTMT Labeling

One μL of 0.5 M DTT was added to 100 μL of BSA while another 100 μL aliquot of BSA was left untouched. Both were then incubated at 37 °C for 1 h. The iodoTMT labels used were each solubilized in 10 μL of methanol and then spun down at 1500 g for 1 min at 4 °C. Then, 5 μL of each label was added to each respective sample, and they were placed in 37 °C incubation for one hour. The labeling reaction was then quenched by adding 4 μL of 0.5 M DTT stock. Next, 100 μL from each sample was combined into one vial, 1.2 mL of −20 °C acetone was added to precipitate the proteins, and the combined samples were held at −20 °C overnight. The sample was then spun down at 10,000× *g* at 4 °C for 10 min. The acetone was decanted, and the pellet was left to dry or accelerated in a lyophilizer. The pellet was then solubilized in 200 μL of 50 mM ammonium bicarbonate and combined with the appropriate amount of trypsin (depending on how much protein one is working with and the recommended amount for that specific trypsin). The sample was left to be enzymatically processed by the trypsin overnight at 37 °C, and then 4 μL of 10% TFA was added to acidify the sample. The sample was lyophilized and held in dry pellet form at −20 °C until mass spectrometry.

### 4.12. Prediction of Unknown Protein Structures

Structural data for proteins that had not been explored using X-ray crystallography were drawn from the DeepMinds AlphaFold database [36]. Only protein structures with

high predicted accuracy were utilized, and analysis conducted using these structures considered the predictive nature of the structures in question.

## 5. Conclusions

Phosphine–borane complexes have demonstrated neuroprotection in multiple models of neuronal injury. Elucidating their currently unknown mechanism of action could provide the foundation to design more specific and effective therapeutics. The tandem mass tagging and mass spectrometry-based approach described herein identified several disulfide targets reduced with PB1 treatment in 661W cells. Several of these targets participate in cellular processes associated with apoptosis and neurodegeneration, highlighting the role that phosphine–borane reducing agents such as PB1 could play in neuroprotection. Further work to understand the role of specific reduction targets within key proteins could yield novel phosphine–borane complexes with increased efficiency and specificity.

**Supplementary Materials:** The following supporting information can be downloaded at: https://www.mdpi.com/article/10.3390/inorganics11070310/s1, Table S1: Comparison of PB1- and TCEP-based on molecular characteristics calculated using Molecular Operating Environment software. Figure S1: Summary of major changes between prototype phosphine TCEP and phosphine–borane complex PB1. Figure S2: Computational images showing labeled elongation factors. Figure S3: Computational images showing labeled ubiquitin-related proteins. Figure S4: Computational images showing labeled 60S ribosomal subunit proteins. Figure S5: Computational images showing labeled disulfide-forming cysteines. Figure S6: Computational images showing additional labeled proteins.

**Author Contributions:** Conceptualization, Y.M., R.E.G. and L.A.L.; methodology, Y.M.; formal analysis, Y.M. and L.A.L.; writing—original draft preparation, Y.M.; writing—review and editing, Y.M. and L.A.L.; visualization, Y.M. and L.A.L.; supervision, L.A.L.; project administration, L.A.L.; funding acquisition, L.A.L. All authors have read and agreed to the published version of the manuscript.

**Funding:** This research was supported by funding from Canada Research Chairs, Canadian Institutes for Health Research (PJT-162396), and the National Eye Institute of the National Institutes of Health (R21EY025074).

**Data Availability Statement:** The data presented in this study are available in the article and supplementary material.

**Conflicts of Interest:** LAL is a consultant to Annexon, Prilenia, Janssen, Roche, Perfuse, Genentech, UNITY, Eyevensys, and Santen; is a member of the Scientific Advisory Board for the Gilbert Family Foundation and the Steering Committee for the National Eye Institute Audacious Goal Initiative; and holds patents assigned to the Wisconsin Alumni Research Foundation. The funders had no role in the design of this study; in the collection, analyses, or interpretation of data; in the writing of this manuscript; or in the decision to publish the results.

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
