# Peer review of "Redox Targets for Phosphine–Boranes"

_inorganics, doi:10.3390/inorganics11070310_

Round 1

Reviewer 1 Report

The processes of metabolism, respiration, putrefaction, fermentation, photosynthesis are redox processes. For many years, the biological role of oxidants formed in cells was seen only in their toxic effect. With an increase in the level of oxidants in the body, the development of many human diseases is associated, including atherosclerosis, liver cirrhosis, cataracts, arthritis, coronary heart disease, bronchial asthma, hepatitis, and diabetes. At the same time, pathological changes in cells can be induced not only with an increase in the concentration of oxidizing agents, but also with an increase in the concentration of reducing agents or antioxidants. Numerous studies point to the key role in maintaining the vital activity of cells of the redox state of the extracellular and intracellular environment, which characterizes a certain balance between electron donors and acceptors (or reducing agents and oxidizing agents).

Thiol-containing components are central participants in many biochemical reactions. However, it should be noted that in most cases cysteines do not directly react with hydroperoxides at physiological pH. The tripeptide glutathione, containing glutamic acid, cysteine, and glycine, is one of the main participants in intracellular redox processes. The concentration of glutathione in tissue cells of humans, animals, plants and in aerobic bacteria reaches millimolar values. It has been shown that glutathione is involved in the processes of antioxidant defense, metabolism of xenobiotics and eicosanoids, in the regulation of the cell cycle and gene expression.

The paper presents a new method for identifying the formation of disulfides and the reduction of sulfhydryl groups in protein targets and the effect of new phosphine-borane complexes on these processes, in particular, in preventing neuronal death. The effect was to reduce the content of cysteine through a new redox agent bis(3-propionic acid methyl ester) phenylphosphine-borane complex (PB1) on 661W cell culture. The presented novel solution uses sensitive mass spectrometry as detection and iodine-labeled groups of free cysteines in various reduction states to identify disulfide targets. This method measures specific disulfide reduction targets across the entire cell proteome. In addition, the multiplex capability of the method allows quantitative comparison of different agents within a single run on a mass spectrometer. These features make the proposed method ideal for studying the reductive behavior of phosphine-borane complexes and elucidating the cellular mechanics underlying their neuroprotective action of the proposed phosphine-borane complex. Another feature of the proposed method is the simultaneous incubation with 3-indole acrylic acid matrix at the time of cell lysis instead of delayed incubation with IAA >1 hour after cell lysis and cell washing in the parallel developed OxiTMT method. Mandatory incubation prior to measurement may result in reduced sensitivity and specificity for detecting the effect of short exposure to a redox drug. Another difference is the use of one iodoTMT label instead of two under OxiTMT method conditions, doubling the number of potential multiplexed conditions.

The undeniable advantage of this work was that the authors of the article tested their method on a fairly large number of protein targets, revealing the features of such interactions. Were identified and quantified 1073 unique protein fragments from 628 unique proteins, of which 13 were found to be statistically significant compared to control cells.

It should be noted that the authors should undoubtedly continue their work in this direction in order to gain a more detailed understanding of the mechanism of action of the phenylphosphine-borane complex in order to make these compounds the basis for the development of more specific and effective therapeutic agents.

The above wishes to the authors of the article do not detract from the dignity of this work. The data obtained by the authors will certainly be interesting and useful to researchers.

Typos:

1. line 8: it seems that 2 should be 5

2. line 67: Schema --> Scheme

 Please add citations to the Introduction:

  • 10.1007/s00775-022-01937-4
  •  

10.3390/molecules25040828

https://doi.org/10.1016/j.ccr.2020.213684

Reviewer 2 Report

In this manuscript, the authors developed a new method for disulfide formation as well as its reduction in proteins targets of redox-active drugs. They verified the validity of the method by using it to elucidate the role of a phosphine-borane complex (PB1) in preventing neuronal death. I would recommend acceptance after the following point is addressed.

1. Concentration-dependence and cytotoxicity: The concentration-dependence of PB1 will give a clear picture about the kinetics/mechanism of action. In addition, the authors should elaborate about the side effects/side reactions of this phosphine-borane complex.

English language is fine. Minor editing is required.
